# The Impact of Work-Related Barriers on Job Satisfaction of Practitioners Working with Migrants

Hannah Brendel, Maha Yomn Sbaa, Salvatore Zappala *, Gabriele Puzzo and Luca Pietrantoni *

Department of Psychology, University of Bologna, Via Berti Pichat 5, 40127 Bologna, Italy
* Correspondence: salvatore.zappala@unibo.it (S.Z.); luca.pietrantoni@unibo.it (L.P.)

**Abstract:** The work environment of practitioners working with migrants may be very demanding as they are frequently exposed to the sad narratives of such a vulnerable population, the lack of professional support, or the frequent change of policies towards refugees and asylum seekers. Little research has been conducted to explore the job satisfaction of practitioners working with migrants and the organizational characteristics that can hinder or promote such satisfaction. The present study investigated the relationship between work-related barriers (i.e., intra-organizational, legal, and interaction-related barriers) and job satisfaction of practitioners working with migrants, also testing if perceived organizational efficacy is mediating this relation. This study was part of a larger European funded project, and participants were 428 First-Line Practitioners working with migrants in various sectors (e.g., social and health services, immigration and asylum services, or border guards) and working in several European countries. Data were collected through an online survey in the period between October and December 2020. Results showed that intra-organizational and legal barriers had a negative impact on job satisfaction, while interaction-related barriers did not have any. Perceived organizational efficacy mediated the relationship between two work-related barriers (intra-organizational and interaction-related barriers) and job satisfaction. These findings suggest that organizations working with migrants should focus on addressing intra-organizational and legal barriers, and on implementing actions aimed at building employees' collective efficacy beliefs to improve their job satisfaction.

**Keywords:** organizational efficacy; job satisfaction; migration; First-Line Practitioners; barriers





## 1. Introduction

Over the past decades, Europe has had, and will have in the future, a key importance as a destination for immigration (Eurostat 2022). Each migration flow and its specific circumstances are unique, constituted by multiple factors such as the drivers for migration, migration route, demographics of migrants or refugees, their countries of origin, and their settlement pattern (Crawley et al. 2018). While the academic literature to date has revealed the challenge of yielding concise definitions and categories for migrants (Bayerl et al. 2020), in the current paper, migrants are those individuals who cross international borders and change their habitual place of residence for a period of at least 12 months, so that the country of destination effectively becomes the individual's new place of habitual residence (Seiger et al. 2022). Migrants comprise refugees, asylum seekers, economic migrants, displaced persons, and individuals migrating for other reasons, such as family reunification.

Organizations in Europe provide help to the abovementioned typologies of migrants upon their arrival in a European country. Such help is oriented towards providing migrants with first support and relief from the travel, and/or information about legal or social procedures useful for integration in the foreign society where migrants just arrived. First-Line Practitioners (FLPs) and their governmental, non-governmental, or intergovernmental organizations provide that help to migrants. The help operates in different sectors, such as rescue at sea, emergency relief, and first reception, as well as legal aid and assistance in

administrative procedures. For instance, lawyers or legal organizations provide refugees and asylum seekers with (pro bono) legal counseling, support in asylum procedures, and provide information or practical support for obtaining access to social and healthcare services. Other FLPs, and their organizations, provide services for longer-term integration of migrants into society as well as for access to social services (e.g., health and psychological, housing, child services, migrant integration, education, or inclusion in the labor market). Moreover, the protection of fundamental rights, promotion of inclusion in the receiving society, and training and mentoring are other areas in which other types of FLPs work with migrants (European Economic and Social Committee 2017).

The work environment of FLPs working with migrants can be very demanding. Particularly, health and social workers are exposed to the narratives of migrants, refugees, and asylum seekers that often report of premigration traumas (due to experience of, for instance, violence, war, torture, or sexual abuse), which are mostly accompanied by postmigration stressors upon their arrival in Europe (cross-cultural stress while integrating into host society, loss of social networks, etc.) (Robertshaw et al. 2017). However, humanitarian workers come from different educational backgrounds and thus have often not received the kind of training that is part of social workers' or psychologists' education, such as in clinical supervision and the understanding of trauma and human development. This, therefore, might lead not psychologically trained workers to higher somatic and burnout symptoms in comparison to psychologically trained workers (Pell 2013). A sample of German refugee aid workers expressed that learning about mental health problems and how to intervene was more important to them than learning about self-care and receiving psychological support for themselves (Grimm et al. 2017). Moreover, evidence suggests that humanitarian work can cause secondary traumatic stress (i.e., secondary exposure to trauma), vicarious trauma, and burnout in practitioners (Guskovict and Potocky 2018; Roden-Foreman et al. 2017). Interestingly, symptoms of secondary traumatic stress and high levels of compassion satisfaction (i.e., satisfaction that results from an individual's ability to aid) have also been demonstrated to coexist (Lusk and Terrazas 2015; Stamm 2010).

A systematic review conducted by Robertshaw et al. (2017) discusses factors that help or hinder health professionals in providing primary healthcare for refugees and asylum seekers, which they refer to as challenges and facilitators. Challenges comprise communication difficulties (e.g., language barriers), cultural understanding (e.g., different cultural values and understanding of the healthcare system), increased time expenditure (e.g., increased duration/occurrences of appointments), lack of training and professional support, increased costs, workforce shortages, and policy restrictions hindering the meeting of health needs and frequently changing policies towards refugees and asylum seekers. Such challenges may hinder health professionals' work with refugees and asylum seekers and thus may impede healthcare organizations to effectively provide primary healthcare for refugees and asylum seekers. In the present study, the above-listed challenges are referred to as work-related barriers, which are here defined as work-related variables that inhibit an organization's effectiveness in working with migrants.

Although the literature mainly focuses on barriers experienced by health and social professionals working with migrants, some barriers are assumed to be generalizable to other work areas of FLPs working with migrants (e.g., border enforcement, immigration, and asylum services), such as frequently changing policies, communication and cultural barriers, and a lack of training for working with such a vulnerable population.

The above-described work environment of practitioners working with migrants and the challenges with which these practitioners are confronted represent barriers that may not only hinder them in providing quality aid services (Robertshaw et al. 2017) but also decrease their job satisfaction. The relationship between organizational barriers, challenges and drivers, and job satisfaction has been well documented in the literature (Judge et al. 2020; Wyrwa and Kaźmierczyk 2020), and although stress and burnout (Nonnis et al. 2020) or quality of life (Mavratza et al. 2021) of FLPs working with migrants have been recently investigated, limited attention has been given to their job satisfaction.

In addition, repeatedly experiencing the presence of barriers might suggest that the organization is less effective in managing such barriers which, in turn, might lead to a decreased job satisfaction. Thus, the present study aims to test if work-related barriers are related to job satisfaction, and if such relationship is mediated by perceived organizational efficacy, in a sample of FLPs working in different Europe-based organizations that provide help and support to migrants. Such an investigation of underlying mechanisms regarding the relationship between work-related barriers and job satisfaction could be very enriching to understand how or why specific work-related barriers are negatively related to job satisfaction. Correspondingly, this could have important implications for organizations in this field, such as allowing for the design of interventions that aim at promoting collective efficacy beliefs in practitioners.

*1.1. Work-Related Barriers and Job Satisfaction*

Job satisfaction is one of the most extensively investigated job attitudes as it can affect employees' wellbeing as well as a variety of behaviors within an organization (George and Jones 2008). Job satisfaction is a multidimensional construct and has been defined in various ways. Locke (1976) defined job satisfaction as "a pleasurable emotional state resulting from the perception of one's work as fulfilling or enabling the fulfillment of significant values available at work, provided that these values are convergent with one's needs" (p. 1319). A systematic literature review conducted by Wyrwa and Kaźmierczyk (2020) aimed at analyzing and synthesizing the variety of definitions of job satisfaction and its determinants based on publications in the fields of psychology, sociology, economics, and management science between 2000 and 2018. Based on their review, prior research investigating the antecedents of job satisfaction emphasized objective and subjective factors as important. Objective factors are shaped by employees and are directly connected to the work environment, comprising job content (such as autonomy or task diversity) and job conditions (such as pay or promotion). Moreover, it is stated that an individual's level of job satisfaction is strongly influenced by situational factors within the field of occupation, followed by a medium influence of personality factors, and finally by the interaction between those two types of factors.

Situational factors can affect job satisfaction both positively and negatively. If such factors are well managed, they have a positive effect; if badly managed or coped with, they may turn into barriers in the workplace and have a negative effect on job satisfaction. Situational factors that should be considered when assessing job satisfaction in the organizational context are job characteristics, job conditions, work climate, organizational culture, quality of organizational management, workload in hours, stress level, colleagues and supervisors, pay and reward, conflict, and the labor market situation (Wyrwa and Kaźmierczyk 2020). A previous study exploring situational determinants of job satisfaction has demonstrated that job conditions, type and nature of tasks, organizational culture, promotion, and pay have a significant effect on job satisfaction (Judge et al. 2009).

Situational factors that are present in the work environment of practitioners working with migrants may shape their level of job satisfaction as well. Communication difficulties, lack of training and professional support, policies towards refugees and asylum seekers, exposure to narratives of a vulnerable population, and stress caused by the work performed may be perceived by practitioners as work-related barriers, influencing their job satisfaction negatively. However, the existence of such work-related barriers may not only have a negative effect on employees' wellbeing and job satisfaction, but may also affect employee behavior and motivation (e.g., employee withdrawal behavior) as well as an organization's operations (Hardy et al. 2003; Henne and Locke 1985; Mobley 1977; Wyrwa and Kaźmierczyk 2020). Correspondingly, barriers of FLPs working with migrants could lead to declines in the quality of provided aid services.

Only a few studies have investigated the work environment of FLPs working with migrants, and how it may affect employees' job attitudes. Moreover, existing research has primarily been conducted among social and healthcare workers, mostly not considering

practitioners working in sectors such as border or customs enforcement, governance and policymaking, or legal services (e.g., Robertshaw et al. 2017; Robinson 2013). Thus, the current study aims to bridge this gap by investigating the effect work-related barriers may have on job satisfaction in FLPs working with migrants. The work-related barriers that are being investigated in the present study include, amongst others, legal constraints, insufficient human resources, lack of training, insufficient salary, lack of coherent strategies and procedures, and language and cultural barriers. Further, workers from many sectors are also considered (e.g., immigration advocacy, immigration and asylum services, legal services, diplomatic sector, governance and policymaking, and customs and internal law enforcement).

Based on the presented findings on the influence of situational factors on job satisfaction, the following assumption can be made: if FLPs working with migrants perceive work-related barriers in working with migrants, they will be less satisfied with their job (Figure 1 describes the whole model hypothesized in this study). Therefore, it is hypothesized that:

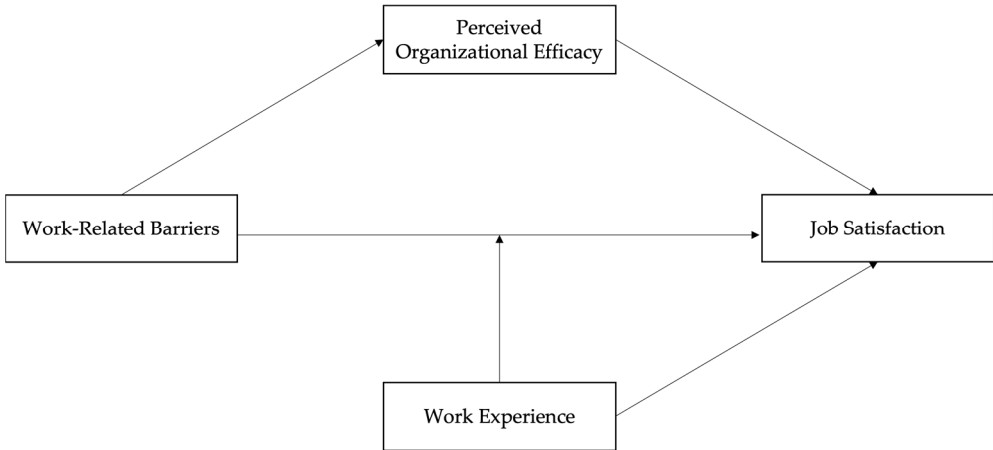

**Figure 1.** Hypothesized Model.

**H1:** *Work-related barriers are negatively related to job satisfaction.*

*1.2. Perceived Organizational Efficacy as a Mediator*

As mentioned, practitioners working with migrants are typically faced with several barriers within the organizational context. Such barriers may be uncontrollable in nature through personal agency (e.g., legal constraints, frequently changing policies) or solely controllable through the exercise of proxy agency, that is by trying to get other individuals with expertise or power to achieve desirable outcomes (e.g., lack of professional training, insufficient salary). This may lead practitioners to experience failure. An individual's perception of one's performance being a failure may lower the individual's efficacy beliefs and may lead an individual to expect future performances to be unsuccessful as well (Goddard et al. 2004). Applying this to the organizational level, FLPs may experience successes and failures as a group. Past successes of an organization may contribute to shaping FLPs' beliefs in the capability of their organization in performing productively, whereas past failures may lead to a decreased sense of organizational efficacy. Thus, according to the social cognitive theory (Bandura 1997), such experiences may contribute to shaping collective efficacy beliefs. For that reason, the current research aims to investigate the effect that context variables (i.e., work-related barriers) may have in shaping collective efficacy beliefs more deeply.

Prior research investigating the construct of collective efficacy has mostly focused on the team level and only few have investigated it on the organizational level. However, knowledge on perceived organizational efficacy could be very enriching and would allow to draw implications for the comprehension, assessment, and generation of collective

efficacy in such larger collectives (Zaccaro et al. 1995). Further, previous studies have mostly considered the perception of employees regarding colleagues, direct supervisors, or top management (Borgogni et al. 2009, 2010; Petitta and Borgogni 2011). Results of such studies suggest that perceptions of the organizational context are relevant dimensions which contribute to shaping collective efficacy (Borgogni et al. 2010). In line with this, it can be assumed that organizational contextual variables (i.e., barriers) may hinder the development of an employee's organizational efficacy beliefs.

It has been suggested that organizational efficacy beliefs are related to various work outcomes, such as organizational commitment and job satisfaction (Borgogni et al. 2010; Kravchenko and Zappalà 2017). This happens because, when employees share the perception that their organization is capable of effectively managing the routinary and less routinary events that arise, their feeling of belonging to a valued and worthy organization is reinforced. This resulting positive work environment thus represents an auxiliary condition for an employee to persevere and manage their own tasks when being confronted with work-related barriers. According to the social cognitive theory, individuals' beliefs in the organization's capabilities of performing productively (i.e., collective efficacy) can positively affect group members' individual and collective motivation and behaviors and increase the group's perseverance when difficulties arise (Bandura 2000). In the opposite, perception of organizational inefficacy in solving problems can increase the belief that employees' efforts will be in vain because the organization is unable to coordinate and combine the efforts of all its components, thus decreasing employees' satisfaction for the job they have done.

The current research aims to expand the knowledge on perceived collective efficacy as a mediator explaining the relationship between contextual variables (i.e., work-related barriers) and job satisfaction. Considering antecedents and consequences of collective efficacy beliefs, the following can be assumed: If FLPs working with migrants perceive work-related barriers, employees' beliefs in their organization's capabilities in performing productively (i.e., perceived organizational efficacy) may be reduced. This may result in FLPs working with migrants to be less satisfied with their job. Therefore, it is hypothesized that:

**H2:** *Perceived organizational efficacy mediates the relationship between work-related barriers and job satisfaction.*

### 1.3. The Role of Work Experience

Work experience has been conceptualized differently in terms of the context to which the work experience relates. Work experience can refer to the accumulated years an individual has been working in one organization (i.e., job tenure), in one specific field, or to the accumulated years an individual has been working in general (possibly across several organizations). In the present study, we conceptualize work experience as the accumulated years FLPs have been working in their current field.

Years of work experience in one specific field represents a commonly investigated construct in relation to job satisfaction. Results of such studies, however, have been inconsistent. For instance, research suggests that years of work experience are unrelated to job satisfaction (Matos et al. 2010) or that length of teaching experience does not significantly affect teachers' job satisfaction and work engagement (Topchyan and Woehler 2021).

On the other side, however, Wyrwa and Kaźmierczyk (2020) report that employees with a higher tenure in an organization show higher levels of job satisfaction. In line with this, prior research has shown that job satisfaction is lower among academic nurses with less than ten years of work experience in the academic field (Altuntaş 2014). Moreover, the scoping review conducted by Wirth et al. (2019) suggests that work experience is a protective factor among employees working with refugees, being associated with higher compassion satisfaction and lower emotional exhaustion. Further, a study conducted by Shaheen et al. (2021) has demonstrated that nurses' job satisfaction increases with the years of experience in the profession. The authors suggested that job satisfaction is higher when employees have more years of work experience within the profession

as employees may receive more recognition for their work, have established better relationships with supervisors, and have developed a strong attachment to their workplace (Shaheen et al. 2021).

Work experience is here suggested to be representing a protective factor for FLPs working with migrants that is maintained even after workers change their workplace and continue to work in another organization in the same field. Drawing on the findings of the relationship between work experience and job satisfaction, we expect that FLPs who have more years of work experience will be more satisfied with their job.

Organizational context variables, such as physical job conditions or pay, may have a differential effect on job satisfaction contingent upon the years of work experience. Following this, Shaheen et al. (2021) called for the necessity of further exploring the relationship between organizational context variables and an individual's work experience to clarify if there is an interaction between these variables and an individual's job satisfaction. Following Shaheen et al.'s (2021) call, the current study aims to investigate the moderator role of work experience in the relationship between organizational context variables (i.e., work-related barriers) and job satisfaction. It is argued here that higher work experience in a specific field might also mean an increase in FLPs' skills and confidence in being able to manage work-related barriers and/or persevere when being confronted with such barriers. As a result, practitioners might experience less failure due to successful management when being confronted with work-related barriers, which might favor a higher level of job satisfaction. Therefore, it is hypothesized that:

**H3:** *Work experience will moderate the relationship between work-related barriers and job satisfaction. More precisely, it is hypothesized that the negative relationship between work-related barriers and job satisfaction will be weaker for practitioners with more years of work experience than for practitioners with fewer years of work experience.*

## 2. Materials and Methods

### 2.1. Procedure

The present study is part of a European Research and Innovation project on security issues and border management. Prior to the launch of the survey, consortium partners of the EU-funded project PERCEPTIONS (constituted by 15 different institutions, based in twelve European and three non-European countries) established a contact list including different associations, institutions, NGOs, and law enforcement agencies based in Europe.

The data were collected through an online survey distributed between October and December 2020. First-Line Practitioners were invited to participate by email and additionally by telephone. At the beginning of the questionnaire, participants had to sign an informed consent form. The questionnaire has been subjected to and approved by the Ethics Committee of the European project PERCEPTIONS.

### 2.2. Measures

Work-Related Barriers. Participants were asked to answer the following question: "To what extent is the following work-related barrier inhibiting your organization's effectiveness in working with migrants?". The list of 14 work-related barriers ($\alpha = 0.90$) was developed during a pilot study in organizations working with migrants. The response format consisted of an 11-point scale, from 0 = not at all to 10 = severely.

Perceived Organizational Efficacy. Perceived organizational efficacy was assessed by two questions ($\alpha = 0.87$). An example is "How effective is your organization in its work with migrants overall?". The response format consisted of an 11-point scale, from 0 = not effective to 10 = extremely effective.

Job Satisfaction. Job satisfaction was measured through five items ($\alpha = 0.84$). Participants were asked to rate how satisfied they were with their current work conditions. The five items were related with satisfaction and with work-life balance, work conditions, salary, social recognition, and the overall satisfaction. The response format consisted of an 11-point scale, ranging from 0 = extremely dissatisfied to 10 = extremely satisfied.

Work Experience. Participants were asked to indicate the number of years they had been working as First-Line Practitioners in organizations providing help to migrants.

### 2.3. Participants

Participants were FLPs in organizations working with migrants in EU countries (i.e., Austria, Belgium, Bulgaria, Cyprus, France, Germany, Greece, Italy, Lithuania, The Netherlands, Spain). The sample consists of 428 participants, from which 55.7% were female ($n$ = 233) and 44.3% male ($n$ = 185). Age ranges were used, and the largest age category was 30–39 years old. Most of the participants work in non-governmental organizations (49.0%, $n$ = 198), followed by governmental organizations (36.9%, $n$ = 149), intergovernmental organizations (4.2%, $n$ = 17), and other organizations (9.9%, $n$ = 40). Respondents belong to many different service sectors, and for our analysis, we grouped the different types of services into five categories. In sector 1, we included FLPs working in immigration advocacy, psychological and health, housing, child, women, and labor services for migrants (47.0%, $n$ = 201); sector 2 included legal services (27.1%, $n$ = 116); sector 3 included FLPs working in border, customs, and law enforcement (17.1%, $n$ = 73); diplomatic and governmental bodies were included in sector 4 (4.4%, $n$ = 19), and, finally, sector 5 included other sectors (4.4%, $n$ = 19).

Participants were excluded when their answers on the dimensions perceived organizational efficacy, work-related barriers, job satisfaction, and work experience contained at least 10% missing values. On that account, 84 participants were excluded.

## 3. Results

### 3.1. Factorial Analysis on Work-Related Barriers

Factorial structure was tested through Exploratory Factor Analysis (EFA), including 14 items measuring work-related barriers. Bartlett's test of sphericity was significant ($\chi^2$ (91) = 2614.84, $p$ < 0.001). The Kaiser–Meyer–Olkin measure of sampling adequacy indicated a high relationship among variables (KMO = 0.89). Further, extraction communalities of all items were above 0.30, confirming that each item shared common variance with the other items. Considering these indicators, factor analysis was suitable with 14 items. Following Kaiser's rule of retaining factors with eigenvalues higher than one, a three-factor solution was suggested, explaining a total of 64.1% of the variance. The scree plot suggests a three-factor solution as well, as eigenvalues "level off" after three factors. Considering the pattern matrix, none of the items were eliminated as all items met the minimum criteria of having a primary factor loading of 0.40 or above and no item showed cross-loadings of 0.30 or above. Within the pattern matrix, all items had primary loadings higher than 0.50.

The first factor ("intra-organizational barriers") consists of nine items and includes barriers related to the organization (e.g., lack of coherent strategies and procedures, scarcity of suitable operational tools) and to the employee (e.g., lack of expertise, burden). The second factor ("legal barriers") consists of two items, including the barriers of legal constraints and jurisdictional conflicts. The third factor includes barriers related to the interaction between FLPs and migrants ("interaction-related barriers") and consists of three items, including language and cultural barriers in the interaction with migrants and scarcity of comprehensive information about the migrants. Internal consistency for each of the subscales was examined using Cronbach's alpha, demonstrating to be acceptable for legal barriers ($\alpha$ = 0.79) and interaction-related barriers ($\alpha$ = 0.80) and good for intra-organizational barriers ($\alpha$ = 0.90).

### 3.2. Correlations for Key Study Variables

Means, standard deviations, internal consistencies, and intercorrelations among all variables are displayed in Table 1.

**Table 1.** Descriptive Statistics and Correlations for Key Study Variables.

|  | M | SD | 1 | 2 | 3 | 4 | 5 | 6 |
|---|---|---|---|---|---|---|---|---|
| 1. Job Satisfaction | 6.17 | 2.00 | (0.84) | | | | | |
| 2. Perceived Organizational Efficacy | 7.71 | 1.87 | 0.20 ** | (0.87) | | | | |
| 3. Intra-Organizational Barriers | 4.76 | 2.35 | −0.24 ** | −0.21 * | (0.90) | | | |
| 4. Legal Barriers | 4.60 | 3.05 | −0.15 ** | −0.08 | 0.43 ** | (0.79) | | |
| 5. Interaction-Related Barriers | 4.23 | 2.69 | −0.06 | −0.22 ** | 0.55 ** | 0.18 ** | (0.80) | |
| 6. Work Experience | 10.18 | 8.05 | 0.12 * | 0.10 * | −0.06 | −0.03 | −0.03 | – |

Note: * $p < 0.05$; ** $p < 0.01$; Reliability coefficients of the scales (Cronbach's $\alpha$) are provided in brackets.

Work experience ranged from 1 to 40 years ($M = 10.18$; $SD = 8.05$). Mean scores for job satisfaction ($M = 6.17$, $SD = 2.00$) and perceived organizational efficacy ($M = 7.71$, $SD = 1.87$) were above the midpoint of 5.0. A one-way ANOVA revealed that there was no statistically significant difference in the level of job satisfaction ($F_{(4,423)} = 0.662$; $p = 0.62$) and perceived organizational efficacy ($F_{(4,420)} = 1.924$; $p = 0.11$) between the different sectors in which FLPs work (i.e., immigration advocacy, psychological and health, housing, child, women, and labor services (sector 1), legal services (sector 2), border, customs, and law enforcement (sector 3), diplomatic and governmental bodies (sector 4), and other sectors). Among work-related barriers, mean scores for intra-organizational ($M = 4.76$, $SD = 2.35$), legal ($M = 4.60$, $SD = 3.05$), and interaction-related barriers ($M = 4.23$, $SD = 2.69$) were slightly below the midpoint of 5.0. All standard deviations are above 1.8, indicating that responses are spread out over a large range of values.

Cohen's guidelines (Cohen 1988) were used for the interpretation of the effect size magnitude, where r = 0.10, r = 0.30, and r = 0.50 indicate small, medium, and large, respectively. Job satisfaction was positively related to perceived organizational efficacy and work experience, and negatively related to intra-organizational and legal barriers. Interestingly, job satisfaction was not related to interaction-related barriers. Perceived organizational efficacy was negatively related to intra-organizational and interaction-related barriers; however, it was unrelated to legal barriers. Further, perceived organizational efficacy was positively related to work experience. All three types of barriers were significantly interrelated with each other. Work experience was unrelated to all three types of work-related barriers.

*3.3. The Effect of Intra-Organizational Barriers on Job Satisfaction*

We verified assumptions for testing the direct and indirect effects as well as the conditional direct effect for the three types of work-related barriers on job satisfaction. Within all analyses for testing the assumptions, collinearity was not existent as tolerance levels were close to 1, the Variation Inflation Factors (VIFs) were all below 10, and Condition Indices were all below 15. Moreover, residuals were proven to be independent as Durbin–Watson (DW) values were in the range 1.5 < DW < 2.5. Lastly, assumptions of homoscedasticity for each of the independent variables and normal distribution of residuals of the independent variables were met.

Results showed that intra-organizational barriers negatively predict perceived organizational efficacy ($\beta = -0.17$, $p < 0.001$), which, in turn, positively predicts job satisfaction ($\beta = 0.14$, $p < 0.01$). The bootstrap confidence interval for the indirect effect of intra-organizational barriers on job satisfaction through perceived organizational efficacy did not include zero ($a_1 b_1 = -0.02$; CI 95% = [−0.052, −0.003]), providing support for Hypothesis 2. Therefore, it can be concluded that perceived organizational efficacy mediated the relationship between intra-organizational barriers and job satisfaction. Moreover, intra-organizational barriers had a significant and negative direct effect on job satisfaction ($\beta = -0.18$, $p < 0.001$), rendering support for Hypothesis 1. Thus, practitioners that perceive a high form of intra-organizational barriers are less satisfied with their job.

Moreover, work experience significantly moderated the direct effect of intra-organizational barriers on job satisfaction ($\beta = 0.01$, $p < 0.05$; CI 95% = [0.002, 0.021]), pro-

viding support for Hypothesis 3. To further interpret the interaction effect between intra-organizational barriers and work experience on job satisfaction, and following Aiken and West (1991), simple slopes for high and low values of the moderator (i.e., one standard deviation above and below the mean) were computed. When work experience was low ($-1$ SD), the slope estimating the relationship between perceived intra-organizational barriers and job satisfaction was statistically significant and negative ($\beta = -0.27$, $p < 0.001$). When work experience was high ($+1$ SD), this slope was non-significant ($\beta = -0.08$, $p > 0.05$). Lastly, the overall conceptual model was significant, explaining about 9.9% of the variance in job satisfaction, $F$ (4,420) = 11.58, $p < 0.001$, $R^2 = 0.099$.

### 3.4. The Effect of Legal Barriers on Job Satisfaction

Results showed that legal barriers do not significantly predict perceived organizational efficacy ($\beta = -0.05$, $p > 0.05$). Perceived organizational efficacy, however, positively predicts job satisfaction ($\beta = 0.19$, $p < 0.001$). Further, the bootstrap Confidence Interval for the indirect effect of legal barriers on job satisfaction through perceived organizational efficacy included zero ($a_1b_1 = -0.01$; CI 95% = [$-0.026$, 0.001]), leading to the rejection of Hypothesis 2. Thus, perceived organizational efficacy did not mediate the relationship between legal barriers and job satisfaction.

Legal barriers, however, had a significant direct effect on job satisfaction ($\beta = -0.09$, $p < 0.01$), providing support for Hypothesis 1. Therefore, practitioners that perceive a high form of legal barriers are less satisfied with their job. Moreover, work experience did not significantly moderate the effect of legal barriers on job satisfaction ($\beta = 0.004$, $p > 0.05$; CI 95% = [$-0.004$, 0.012]), leading to the rejection of Hypothesis 3. Lastly, the overall conceptual model was significant, explaining about 7.0% of the variance in job satisfaction, $F$ (4,412) = 7.72, $p < 0.001$, $R^2 = 0.070$.

### 3.5. The Effect of Interaction-Related Barriers on Job Satisfaction

Results showed that interaction-related barriers negatively predict perceived organizational efficacy ($\beta = -0.15$, $p < 0.001$), which, in turn, positively predicts job satisfaction ($\beta = 0.18$, $p < 0.001$). The bootstrap Confidence Interval for the indirect effect of interaction-related barriers on job satisfaction through perceived organizational efficacy did not include zero ($a_1b_1 = -0.03$; CI 95% = [$-0.055$, $-0.006$]), providing support for Hypothesis 2. Thus, it can be concluded that perceived organizational efficacy mediated the relationship between interaction-related barriers and job satisfaction. Furthermore, interaction-related barriers did not have a significant effect on job satisfaction ($\beta = -0.02$, $p > 0.05$), leading to the rejection of Hypothesis 1. Thus, it can be concluded that perceived organizational efficacy fully mediates the relationship between interaction-related barriers and job satisfaction.

Further, work experience did not significantly moderate the effect of interaction-related barriers on job satisfaction ($\beta = 0.007$, $p > 0.05$; CI 95% = [$-0.001$, 0.015]), leading to the rejection of Hypothesis 3. Lastly, the overall conceptual model was significant, explaining about 5.5% of the variance in job satisfaction, $F$ (4,420) = 6.16, $p < 0.001$, $R^2 = 0.055$.

## 4. Discussion

The present study addressed the role of work-related barriers (i.e., intra-organizational, legal, and interaction-related barriers) in determining job satisfaction within practitioners working with migrants. The work environment of FLPs working with migrants may be very demanding due to exposure to the narratives of such a vulnerable population, insufficient training, lack of professional support, or frequently changing policies towards refugees and asylum seekers (Guhan and Liebling-Kalifani 2011; Robertshaw et al. 2017). However, only a few studies investigated the work environment of FLPs working with migrants and its effect on employees' job satisfaction. To provide a more differentiated insight into the relation between a demanding work environment and job satisfaction, different types of work-related barriers and their relation to job satisfaction were investigated in the present study.

Results indicated that intra-organizational barriers and legal barriers negatively predicted job satisfaction. However, this was not the case for interaction-related barriers. Thus, Hypothesis 1 was confirmed for intra-organizational and legal barriers, but not for interaction-related barriers. In general, situational factors within the field of occupation may affect job satisfaction positively or negatively (Wyrwa and Kaźmierczyk 2020). If such situational factors are badly managed or coped with, they might turn into work-related barriers, having a negative effect on job satisfaction. The results of the present study show that the situational factors such as a lack of coherent strategies and procedures, poor coordination among stakeholders, insufficient human resources (constituting intra-organizational barriers), and legal constraints and jurisdictional conflicts (constituting legal barriers) contribute in shaping FLPs' job satisfaction. However, the question of why interaction-related barriers do not contribute to shaping FLPs' job satisfaction needs to be raised. In the present research, interaction-related barriers consisted of language barriers, cultural barriers, and a lack of comprehensive data on migrants. It is here argued that possibly speaking a different language or having different cultural backgrounds can be considered as integral parts and foreseeable characteristics when working with migrants. Moreover, a variety of studies report that working in the refugee sector strongly supports workers' values, beliefs, and interests and that workers may derive strong meaning from their work with migrants (Guhan and Liebling-Kalifani 2011; Robinson 2014). It is here argued that these aspects and FLPs' apparent high intrinsic motivation to perform their job may outweigh the difficulties practitioners might experience due to language and cultural barriers and could serve as an explanation for why interaction-related barriers do not contribute to shaping job satisfaction. Another reason for this could be that practitioners may have developed suitable coping mechanisms supporting them to overcome or manage such interaction-related barriers.

The present study investigated the mediating role of perceived organizational efficacy in the relationship between the three types of work-related barriers and job satisfaction. Results suggest that perceived organizational efficacy mediates the relationship between intra-organizational barriers and job satisfaction as well as between interaction-related barriers and job satisfaction, rendering support for Hypothesis 2. However, perceived organizational efficacy does not mediate the relationship between legal barriers and job satisfaction, as legal barriers did not predict perceived organizational efficacy. These results are partially in line with previous findings of Borgogni et al. (2010). In the present study, the mediating effect of efficacy beliefs within the relationship between the context variable of work-related barriers (except for legal barriers) and job satisfaction was confirmed.

Results can be explained in light of the social cognitive theory, which suggests that collective and self-efficacy beliefs are shaped when individuals weigh and interpret efficacy belief-shaping information (e.g., direct experiences of failure or success, group mastery experiences, or psychological state) in a specific social context (Bandura 1997; Goddard et al. 2004). It was here suggested that existing barriers with which FLPs are being confronted may result in employees experiencing failure as a group or prohibit group mastery experiences. Such past failures may result in individuals expecting future group performances to be unsuccessful as well and, as a result, contribute to shaping FLPs' beliefs in their organization's capability in performing productively negatively, leading to a lower form of perceived organizational efficacy.

It may be argued, however, that this might only be the case for barriers that are controllable in nature, either through personal or proxy agency. Intra-organizational or interaction-related barriers might be perceived as within the control or capacity of individuals, meaning that the individuals believe that the organization where they work is potentially capable of organizing and executing actions to obtain desired outcomes as well as managing the barriers with which they are being confronted. Regarding legal barriers, however, this might not be the case, as legal constraints and jurisdictional conflicts lay outside their scope of control. Consequently, it is argued that only experiences of failure re-

sulting from work-related barriers that are perceived to be potentially controllable (through personal or proxy agency) contribute to shaping perceived collective efficacy beliefs.

Also, results suggest that perceived organizational efficacy positively predicts FLPs' job satisfaction in the three models that were tested. These results support the effect of perceived organizational efficacy on job satisfaction that has been found in the educational and the military setting (Borgogni et al. 2010; Buonomo et al. 2020; Caprara et al. 2003), and thus demonstrate the generalizability of the effect to the context of FLPs working with migrants. Perceived collective efficacy can positively affect group members' individual and collective motivation and behaviors as well as increase the collective's perseverance when being confronted with difficulties (Bandura 2000). Consequently, collective efficacy beliefs may influence various work outcomes, including job satisfaction.

Additionally, the moderating role of work experience in the relationship between the three types of work-related barriers and job satisfaction was investigated. Results indicated that solely intra-organizational barriers have a differential effect on job satisfaction contingent upon the years of work experience, thus providing partial support for Hypothesis 3. More precisely, intra-organizational barriers negatively predict job satisfaction only in practitioners with low work experience (in years). This means that the level of job satisfaction for FLPs with high work experience in their field does not depend on whether they perceive intra-organizational barriers or not. Thus, it seems that, as Shaheen et al. (2021) suggested, organizational context variables may have a differential effect on job satisfaction contingent upon the years of work experience.

A possible explanation for this finding could be that practitioners with limited work experience in the field have not yet acquired and developed necessary skills, knowledge, or strategies to overcome barriers with which they are being confronted and, consequently, may be more likely to experience failure in comparison to experienced workers. Also, more experienced workers may have learned over the course of time which barriers they themselves are able to manage through personal agency and which barriers can only be managed through proxy agency (i.e., trying to get other individuals with the necessary expertise or power to achieve a desirable outcome). The abovementioned aspects may lead to more experienced versus less experienced employees perceiving failures resulting from intra-organizational barriers differently. This may be explained in the light of attribution theory. The theory suggests that the attribution of specific causes to behavior or outcomes is seen as a process in which an individual collects and assesses information concerning potential causes and in a second step utilizes this information to make inferences regarding a person's dispositions (internal attribution) or characteristics of the environment or situation (external attribution) (Ross 1977). Considering attribution theory, it is here argued that more experienced practitioners may more likely be able to correctly identify when failure resulting from intra-organizational barriers should be attributed externally versus internally. Less experienced practitioners instead may be more likely to erroneously attribute such failures to personal factors compared to environmental or situational factors.

This study provided important findings about work-related barriers practitioners working with migrants perceive and their effect on job satisfaction. However, the study also has some limitations. First, a longitudinal design would better consider the implications of work-related barriers on perceived organizational efficacy beliefs and job satisfaction. Second, perceived organizational efficacy was measured with only two items. In future investigations, collective organizational efficacy may be measured by existing, validated scales (e.g., Bohn 2010) to allow a more complete assessment of perceived organizational efficacy. Third, it would be interesting to consider FLPs' self-efficacy beliefs in future research. This would allow to shed light on the extent to which job satisfaction is explained by collective versus self-efficacy beliefs.

## 5. Conclusions

As the literature on work environment of FLPs and the effects it has on work outcomes is still very sparse, this study extends our knowledge by exploring the barriers that practi-

tioners working with migrants frequently face at their workplace. By following the contextualist theory of knowledge (McGuire 1983), this study showed that intra-organizational and legal barriers, but not interaction-related barriers, negatively affect FLPs' job satisfaction, thus partially supporting Hypothesis 1. In particular, it was observed that, especially for FLPs with less experience in the field, intra-organizational barriers have a negative effect on job satisfaction. Further, in regard to Hypothesis 2, results of this study showed that intra-organizational and interaction-related barriers may have a negative impact on FLPs' perceived collective efficacy, which in turn leads to a lower job satisfaction. Perceived organizational efficacy, however, does not seem to mediate the relationship between legal barriers and job satisfaction. Moreover, the study showed that the negative relationship between intra-organizational barriers and job satisfaction was weaker for employees with more years of work experience than for employees with fewer years of work experience, thus providing partial support for Hypothesis 3.

The findings presented in the current study suggest the significance of organizational actions aimed at building practitioners' collective efficacy beliefs, which, consequently, may increase satisfaction with their job. This could be achieved by providing FLPs with training focusing on skill development, team building to help them understand each other's strengths and weaknesses and build a sense of shared responsibility, or on effective communication (both with their colleagues and with migrants). This could increase FLPs' understanding of how to effectively collaborate with colleagues and coordinate their efforts, resolve conflicts, build trust, and manage emotions. Further, organizational actions should be aimed at lowering the intra-organizational and legal barriers FLPs are experiencing to increase their job satisfaction. The experience of intra-organizational barriers in working with migrants may be mitigated, for instance, by establishing a formal coordination mechanism with stakeholders, including regular meetings to improve coordination among stakeholders, or by offering employee assistance programs (e.g., counseling) or wellness programs (e.g., stress management or mindfulness workshops) to provide support for the stress and psychological burden experienced by FLPs. Organizations may mitigate experienced legal barriers, instead, by providing FLPs with legal support, or by giving regular updates on relevant laws and regulations to support FLPs' understanding of their legal obligations as well as the legal obligations of their organization. Lastly, measures that are aimed at increasing the level of work experience (e.g., through knowledge sharing with more experienced practitioners or mentoring) may be of high value. By implementing such measures, intra-organizational barriers may decrease their impact on job satisfaction. Taken together, organizations need to consider these factors and their relationships with job satisfaction. Lowering the barriers with which practitioners working with migrants are being confronted, and thus improving job conditions, may have a positive impact on their collective efficacy beliefs as well as their job satisfaction, allowing the provision of quality aid services.

**Author Contributions:** Conceptualization, S.Z., H.B., L.P. and G.P.; methodology, S.Z., G.P. and M.Y.S.; resources, L.P.; data curation, S.Z.; writing—original draft preparation, H.B.; writing—review and editing, H.B., L.P., S.Z. and M.Y.S.; supervision, L.P. and S.Z.; project administration, L.P.; funding acquisition, L.P. All authors have read and agreed to the published version of the manuscript.

**Funding:** This research was funded under the European Union's Horizon 2020 Research and Innovation Programme by the European Commission. Title of the project PERCEPTIONS (Grant Agreement N. 833870).

**Institutional Review Board Statement:** Approval was obtained from the Ethics committee of European project PERCEPTIONS. The procedures used in this study adhere to the tenets of the Declaration of Helsinki.

**Informed Consent Statement:** Informed consent was obtained from all subjects involved in the study.

**Data Availability Statement:** The data that support the findings of this study are available from the corresponding author upon reasonable request.

**Acknowledgments:** The authors thank Isabèl Rodriguez from the University of Valencia for her feedback on the theoretical assumptions of this paper.

**Conflicts of Interest:** The authors declare no conflict of interest.

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
