# Peer review of "The Impact of Work-Related Barriers on Job Satisfaction of Practitioners Working with Migrants"

_socsci, doi:10.3390/socsci12020098_

Round 1
Reviewer 1 Report
This is a well-crafted paper that addresses an important yet under-represented group of workers who support marginalized and traumatized persons. The connection to issues of job satisfaction also add to the novelty of the work. The rest of the paper is also well presented, with a thorough review of the literature, strong measures, and a logical flow of findings that is supported by a good discussion of the work in relation to theory and existing research. This is certainly publishable work. One recommendation is to connect the findings to the practical implications of the work especially in relation to training.
Author Response
Thank you for your comments. Please find enclosed our replies.

Reviewer 2 Report
I felt focus should have been given on the migrants a bit more..for eg. their definitions..who are immigrants..where are they based..what kind of organizations they work for.
Author Response

(The authors gave the same response as above.)

Reviewer 3 Report
An important issue is the introduction the mediator variable "perceived organizational efficacy" in the theoretical model.
Author Response

(The authors gave the same response as above.)

Reviewer 4 Report
The article is interesting and well-written and provides many insights into overcoming organizational barriers to the work of immigration workers. The authors in this regard, in the conclusions should propose what actions could be considered to mitigate these barriers, building on the results of their studies
Author Response

(The authors gave the same response as above.)

Reviewer 5 Report
The article "Impact of Work-Related Barriers on Job Satisfaction in Organizations Working with Migrants" deals with an important topic, which are the barriers to job satisfaction in migrant organizations.
The abstract would have to mention where the research was carried out, in which countries or countries, as well as the types of professionals interviewed. When referring to the parties forming part of a consortium, it would be relevant to mention which consortium this is.
Problematization and literature review is adequate. However, it would be relevant that you place the hypotheses not in the theoretical foundation but in the methodology part, since these will be tested in this study, or presented in a model of analysis of the reality from which the hypotheses derive.
The methodology is clear, explaining what types of questions were asked, but the different professionals are not mentioned in the participants. This part would be important to highlight if the lawyer is more satisfied with the work than the Social Workers.
It clearly presents the results as well as the discussion, but it could be clearer regarding the presented hypotheses, clearly assuming that these are confirmed or informed. That said, the discussion could be more connected with the theoretical framework. The conclusion could be more developed referring to explain if the hypotheses are confirmed or not.
Author Response

(The authors gave the same response as above.)
